# High-performance Kerr microresonator optical parametric oscillator on a silicon chip

Edgar F. Perez[1,2], Grégory Moille [1,2], Xiyuan Lu [1,2], Jordan Stone[1,2], Feng Zhou [1,2] & Kartik Srinivasan [1,2] ✉

Optical parametric oscillation (OPO) is distinguished by its wavelength access, that is, the ability to flexibly generate coherent light at wavelengths that are dramatically different from the pump laser, and in principle bounded solely by energy conservation between the input pump field and the output signal/idler fields. As society adopts advanced tools in quantum information science, metrology, and sensing, microchip OPO may provide an important path for accessing relevant wavelengths. However, a practical source of coherent light should additionally have high conversion efficiency and high output power. Here, we demonstrate a silicon photonics OPO device with unprecedented performance. Our OPO device, based on the third-order ($\chi^{(3)}$) nonlinearity in a silicon nitride microresonator, produces output signal and idler fields widely separated from each other in frequency ( > 150 THz), and exhibits a pump-to-idler conversion efficiency up to 29 % with a corresponding output idler power of > 18 mW on-chip. This performance is achieved by suppressing competitive processes and by strongly overcoupling the output light. This methodology can be readily applied to existing silicon photonics platforms with heterogeneously-integrated pump lasers, enabling flexible coherent light generation across a broad range of wavelengths with high output power and efficiency.

Many applications in quantum information science, metrology, and sensing require access to coherent laser light at a variety of wavelengths, ideally in a chip-integrated format suitable for scalable fabrication and deployment. While integrated photonics lasers are highly developed in the telecommunications band[1], many of the aforementioned technologies operate at other wavelengths. To this end, the extension of heterogeneously integrated lasers to other bands has been pursued, with recent demonstrations at 980 nm[2,3] and 2000 nm[4]. However, wavelength access across the entirety of a broad spectral range would demand the challenging integration of several material platforms. In contrast, table-top nonlinear optics[5,6] is widely used to produce coherent light at wavelengths that are difficult to access through direct laser emission. Processes such as optical harmonic generation, stimulated four-wave mixing, and optical parametric oscillation enable the spectral translation and/or generation of coherent light at wavelength(s) that can differ dramatically from those at the input. The development of high-performance nonlinear integrated photonics platforms[7], when combined with compact lasers, may provide a compelling approach for realizing flexible wavelength access on-chip.

Here, we demonstrate high-performance $\chi^{(3)}$ OPO on a silicon microchip. By suppressing competing nonlinear processes that would otherwise saturate parametric gain and by strongly overcoupling the output mode while retaining high overall $Q$, we simultaneously realize wide spectral separation between the participating modes (signal-idler separation > 150 THz), high conversion efficiency (up to ≈ 29%), and useful output power (up to ≈ 21 mW), a compelling combination of properties that, to the best of our knowledge, has not previously been

[1]Joint Quantum Institute, NIST/University of Maryland, College Park, MD, USA. [2]Microsystems and Nanotechnology Division, National Institute of Standards and Technology, Gaithersburg, MD, USA. ✉e-mail: kartik.srinivasan@nist.gov

simultaneously achieved in on-chip OPO. Our work highlights the potential of OPO in silicon photonics to address many requirements for deployable laser technologies in scientific applications, particularly in light of recent progress on heterogeneous integration of III-V lasers and silicon nonlinear photonics[8].

## Requirements for high performance

In a $\chi^{(3)}$ OPO, pump photons at $\nu_p$ are converted to up-shifted signal photons ($\nu_s$, with $\nu_s > \nu_p$) and down-shifted idler photons ($\nu_i$, with $\nu_i < \nu_p$) that satisfy energy conservation ($2\nu_p = \nu_s + \nu_i$). Appreciable conversion efficiency requires phase-matching, so that $2\beta_p = \beta_s + \beta_i$ where $\beta_{p,s,i}$ is the propagation constant for the pump, signal, and idler modes, respectively. In microring resonators, which have periodic boundary conditions, this phase relationship can be recast as $2m_p = m_s + m_i$ where $m_{p,s,i}$ denotes the azimuthal mode order of the pump, signal, and idler modes, respectively. Finally, OPO has a power threshold, meaning that the cavity modes must have sufficiently low loss rates (high loaded $Q$s) that can be exceeded by the available parametric gain.

While phase- and frequency-matching and high-$Q$ are baseline requirements for OPO, additional requirements are imposed if high-performance OPO is to be achieved. With respect to performance, we focus on the output power and conversion efficiency of wide-bandwidth (signal-idler separation) OPO as our main metrics. Another typical OPO metric is its threshold pump power. While threshold can be important in terms of quantifying resonant enhancement of the nonlinear process, it has a less direct impact on many applications, where the most relevant pump power is determined by the desired output power. To achieve high-performance OPO based on these metrics, first, it is necessary to suppress competitive nonlinear processes that, for example, divert pump energy to the creation of frequency components other than the targeted signal and idler frequencies (Fig. 1(a, b)). In addition, to maximize the conversion efficiency for the output field of interest, optimization of the pump injection into the microring and output extraction from the microring is needed (Fig. 1(c)). We discuss each of these items below, starting with resonator-waveguide coupling.

## Increasing the maximum conversion efficiency

The conversion efficiency for the signal (or idler) is dependent on the coupling regime (e.g., overcoupled/undercoupled) of both the signal (or idler) and the pump[9,10]. As a starting point, we consider a simplified three-mode model in which only the pump, signal, and idler modes are allowed to interact, from which the system's maximum conversion

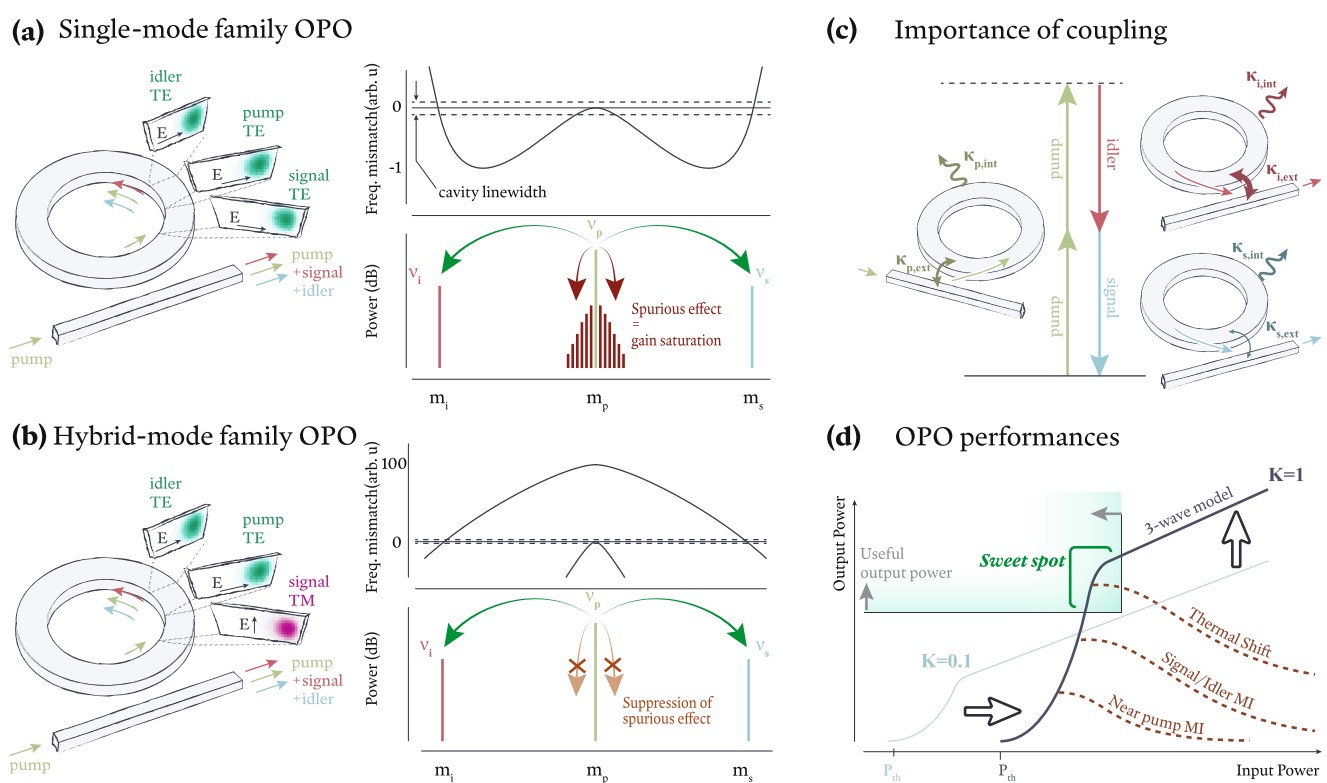

**Fig. 1 | Ingredients for a high-performance $\chi^{(3)}$ microchip OPO. a, b** (Left) Schematic illustration of OPO in two configurations, based on a single mode family (**a**) and a hybrid mode family (**b**) scheme. In each, an input pump (frequency $\nu_p$) is coupled into the microring to generate output signal ($\nu_s$) and idler ($\nu_i$) if the participating modes are phase- and frequency-matched. In **a**, all modes are chosen from the same transverse-electric (TE) mode family (TE$_0$ in this case). In **b**, the modes are chosen from different transverse mode families, including transverse-magnetic (TM) mode families ({TE$_0$, TE$_0$, TM$_0$} for {$\nu_i, \nu_p, \nu_s$} in this case). (Right) Frequency mismatch $\Delta\nu$ (top) and expected OPO spectrum (bottom) for the two approaches. In **a**, frequency matching ($\Delta\nu = 0$) is achieved through a dispersion balance that results in weakly normal dispersion near the pump, which allows for spurious competitive processes that saturate the nonlinear gain. In **b**, the use of mode families with different effective indices enables frequency matching while supporting strong normal dispersion around the pump to largely suppresses

spurious effects (note the difference in y-axis scale between the two cases). **c** The waveguide-resonator coupling must be carefully controlled to optimize conversion efficiency while keeping the operating power within a desirable range. To optimize the conversion efficiency into one output channel (in this case the idler) while maintaining sufficiently high loaded quality factors, it is advantageous to weakly overcouple the input pump ($K_p = \kappa_{p,ext}/\kappa_{p,int} > 1$), strongly overcouple the output idler ($K_i = \kappa_{i,ext}/\kappa_{i,int} \gg 1$), and undercouple the output signal ($K_s = \kappa_{s,ext}/\kappa_{s,int} < 1$), as qualitatively indicated by the weight of the arrows denoting $\kappa_{int}$ and $\kappa_{ext}$ for each mode. **d** OPO output power and conversion efficiency sharply rise once the system crosses threshold, with subsequent growth depending on how well various parasitic processes are suppressed and how well the coupling is engineered. At high enough input powers, the OPO output power will eventually go down once the system becomes frequency mismatched due to Kerr and thermal shifts. Modulation instability is abbreviated as MI.

efficiency, $\eta_{s,i}^{\max} \equiv N_{s,i}/N_p$ can be derived[9]. Here, $N_p$ is the flux of pump photons at the input of the waveguide and $N_{s,i}$ is flux of signal or idler photons at the output of the waveguide. The maximum conversion efficiency, $\eta_{s,i}^{\max}$, will occur when the Kerr-shifted modes are perfectly phase- and frequency-matched, and can be written in terms of the coupling parameter $K_{p,s,i}$ of each resonance as:

$$\eta_{s,i}^{\max} = \frac{1}{2}\frac{K_p K_{s,i}}{\left(K_p+1\right)\left(K_{s,i}+1\right)}, \tag{1}$$

where $K_{p,s,i} = \kappa_{(p,s,i),\mathrm{ext}}/\kappa_{(p,s,i),\mathrm{int}}$ and $\kappa_{(p,s,i),(\mathrm{ext,int})}$ is the extrinsic (waveguide coupling) or intrinsic loss rate for the pump, signal, or idler mode (Fig. 1(c)). This equation shows that $\eta_{s,i}^{max}$ in a $\chi^{(3)}$ OPO increases to a maximum value of 0.5 as $K_{p,s,i}$ increase without bound. However, strongly overcoupling the resonator decreases the total $Q = \nu/(\kappa_{\mathrm{ext}} + \kappa_{\mathrm{int}})$ of the corresponding cavity mode(s), yielding a less efficient nonlinear enhancement. This can translate into very high threshold powers, which may be unsupportable by compact pump lasers. Therefore, efficient OPO generation via overcoupling requires a resonator with very high intrinsic $Q_{\mathrm{int}} \approx \nu/\kappa_{\mathrm{int}}$ as a starting point. In recent years, it has been demonstrated that $Si_3N_4$ microring resonators, suitable for nonlinear photonics and created by mass-production fabrication techniques, can yield intrinsic $Q_{\mathrm{int}} > 10^7$ (Ref. [11]), suggesting that strong overcoupling can be reached while maintaining high overall $Q$.

### Suppressing parasitic processes

In practice, saturation of OPO usually occurs before $\eta_{s,i}^{\max}$ is reached, especially when imposing the additional requirement of achieving $\eta_{s,i}^{\max}$ with high output power. In OPO, the frequency mismatch $\Delta\nu = -2\nu_p + \nu_s + \nu_i$ between the cold-cavity resonances is compensated by their Kerr shifts, which are pump-power dependent quantities, so that there is a limited range of input powers for which $\Delta\nu$ will be small enough for high conversion efficiency to be achieved[9,10]. Thermo-refractive shifts will typically also play a role, and in widely separated OPO the wavelength-dependence of the thermorefractive shifts also becomes meaningful. However, because dispersion is influenced by device geometry[12], these effects can be addressed by choosing a geometry that targets a $\Delta\nu > 0$ compatible with the input power range of interest.

A more significant challenge comes from parasitic nonlinear processes that deplete the gain of the desired OPO process (Fig. 1(a, b)). Competitive parasitic nonlinear processes in this system are a consequence of a microring resonator's many azimuthal spatial modes[10], and can be worsened by the presence of higher-order transverse spatial modes (including those of a different polarization). As a result, in widely separated OPO, there can be hundreds of modes that exist between the pump and targeted signal (or idler) mode, which can be populated by processes such as modulational instability and subsequent Kerr comb formation. These processes are detrimental to system efficiency as they divert pump photons away from the targeted three-mode OPO process. The natural way to limit close-to-pump parasitic nonlinear processes is to situate the pump in the normal dispersion regime, so that Kerr shifts lead to a larger amount of frequency mismatch for nearby signal-idler pairs. However, normal dispersion around the pump (i.e., $\Delta\nu < 0$) must be balanced by sufficient higher-order dispersion for the widely separated signal-idler pair of interest to be frequency matched, so that $\Delta\nu \approx 0$, as in (Fig. 1(a))[9,12,13]. Moreover, the amount of normal dispersion near the pump is also important, as cross-phase modulation involving the widely separated signal and idler modes can result in nonlinear conversion to unwanted spectral channels near the pump if the amount of normal dispersion is insufficient[10]. Thus, a dichotomy arises: strong normal dispersion

suppresses parasitic process, but strong normal dispersion makes the frequency and phase matching conditions challenging to satisfy.

This problem is circumvented though the use of hybrid-mode OPO (hOPO)[14], which phase and frequency matches azimuthal modes from different transverse spatial mode families (Fig. 1(b)). Using this technique, it is possible for each of the pump, signal, and idler bands to have strong normal dispersion, thereby suppressing competitive processes, while maintaining phase and frequency matching for the targeted modes. Hence, through careful design of the resonator's dispersion, it is possible to isolate the hOPO, taking the many-mode system to the limit where it behaves like the modeled three-mode system, where high output power and high conversion efficiency are simultaneously accessible without sacrificing wavelength access.

## Results

### Device design, dispersion, and coupling

Our devices were fabricated by Ligentec (certain commercial products or names are identified to foster understanding. Such identification does not constitute recommendation or endorsement by the National Institute of Standards and Technology, nor is it intended to imply that the products or names identified are necessarily the best available for the purpose) through a photonic damascene process[11] and consist of an $H \approx 890$ nm thick, fully $SiO_2$-clad $Si_3N_4$ microring resonator with outer radius of 23 µm and a ring width of RW $\approx 3$ µm. Figure 2(a) shows a typical cross-section of a microring, which has an inverted trapezoidal shape, with sidewall angle $\theta \approx 16°$ as a result of the reflow step within the damascene process, and whose geometry has been verified through focused ion beam cross-sectional imaging of the devices (see Supplementary Material Fig. S2). Finally, the temperature of the devices was not actively managed; they were passively cooled in an ambient-temperature room.

We first measure the frequency mismatch $\Delta\nu$ for phase-matched sets of signal, idler, and pump modes, as shown in Fig. 2(a) (see Methods). We plot $\Delta\nu$ as a function of the relative mode number $\mu$, indexed with respect to a pump band mode at 308 THz. We consider two cases, the targeted process in which the idler and pump are from the fundamental transverse electric ($TE_0$) mode family and the signal is from the fundamental transverse magnetic mode ($TM_0$) family (shown in purple), and one in which all three modes are from the $TE_0$ mode family (shown in green), i.e., the more typical single mode family case. In the $TE_0$-$TE_0$-$TM_0$ hOPO scheme, $\Delta\nu \approx 0$ at frequencies near 390 THz and 226 THz, indicating that we can anticipate an OPO signal and idler pair near these frequencies for an appropriate level of pump laser detuning and Kerr nonlinear shifts to compensate for any non-zero frequency mismatch. In contrast, $\Delta\nu < 0$ at all frequencies for the $TE_0$-$TE_0$-$TE_0$ case, indicating that the widely separated process of interest will not occur for this set of modes. More importantly, these results confirm that the pump is situated in a regime of normal dispersion, which is explicitly validated through the evaluation of the dispersion parameter $D$ for the $TE_0$ and $TM_0$ mode families, where $D = -\frac{c}{2\pi\lambda^2}\frac{\partial^2\beta}{\partial\nu^2}$. As shown in Fig. 2(b), $D < 0$ for the $TE_0$ family—not only in the pump band, but also in the idler band (as well as the signal band). In addition, $D < 0$ for the $TM_0$ family in the signal band. As discussed above, this normal dispersion throughout the entire frequency range between the signal and idler, and in particular surrounding the pump, should suppress many potentially competing nonlinear processes.

We next move to the resonator-waveguide coupling, which, as noted in a previous section, is critical for a high-performance OPO. We focus on conversion efficiency and output power of the idler generated near 1300 nm. From Eq. (1), $\eta_i^{\max}$ depends on the coupling parameter for the pump $K_p$ and for the idler $K_i$, with greater efficiency being achieved with increased $K_i$ and $K_p$. Additionally, to maintain an acceptable threshold power and because we are not focusing on extraction of the signal, we target small $K_s$. Using a straight waveguide that is at a gap and tangent to the ring naturally leads to variation in

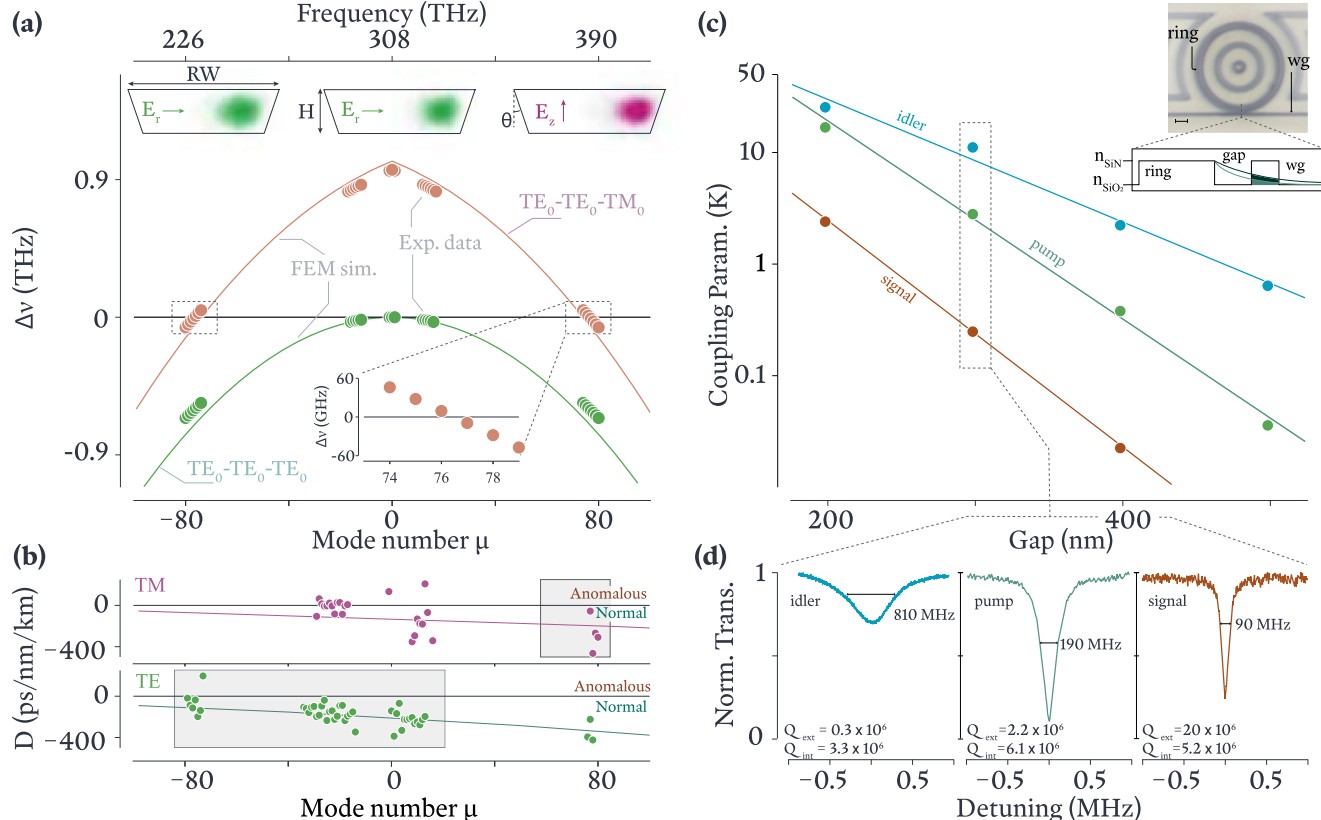

**Fig. 2 | Frequency matching, dispersion, and coupling. a** Frequency mismatch $\Delta\nu = -2\nu_p + \nu_s + \nu_i$ for the hOPO in orange. The solid circles are experimental measurements using a wavemeter (see Methods) to determine the cold-cavity resonance frequencies (uncertainties are within the size of the data points), while the solid curve is from finite-element method (FEM) simulations where the idler, pump, and signal modes are from the $TE_0$, $TE_0$, and $TM_0$ mode families, respectively. The frequency mismatch when all modes are chosen from the $TE_0$ family is shown in the green solid circles and green solid curve for experiment and simulations, respectively. The top images are the simulated transverse electric field profiles for the hOPO modes, where the cross-sectional parameters of ring width (RW), thickness (*H*), and sidewall angle ($\theta$) are indicated. The lower inset magnifies the right zero crossing of the hOPO $\Delta\nu$ curve, showing an $\approx 9$ GHz offset when $\mu = 76$. **b** Dispersion parameter (*D*) for the $TM_0$ and $TE_0$ mode families, with experimental data shown as solid points and fits shown as solid curves. The gray boxes highlight the relevant spectral bands (signal for $TM_0$ and idler and pump for $TE_0$). $D < 0$ for the relevant mode family for each of the idler, pump, and signal bands. **c** Coupling parameter (*K*), defined as the ratio of the resonator-waveguide coupling rate ($\kappa_{ext}$) to the resonator intrinsic loss rate ($\kappa_{int}$), for the idler, pump, and signal modes (blue, green, and orange) as a function of resonator-waveguide gap. The top inset shows an optical micrograph of one device, where the scale bar is 5 μm, along with the refractive index (thin black) profile along a cross-section through the ring and waveguide (wg), and the radial component of the evanescent tail of the idler and pump modes (black and green curves). **d** Transmission spectra for the idler, pump, and signal modes at a gap of 300 nm, along with fitting results for the intrinsic and coupling quality factors.

resonator-waveguide coupling across broad spectral ranges, since the modal overlap between ring and waveguide modes depends on the evanescent decay lengths of each mode (Fig. 2(c) inset), which itself depends on wavelength. As a result, long wavelength modes tend to be overcoupled and short wavelength modes tend to be undercoupled[15], so that $K_i > K_p > K_s$ as desired, provided that intrinsic quality factors remain high throughout. We note that other possible OPO objectives, such as high efficiency and output power for the signal field, or equal efficiency and output power for both signal and idler fields, may require alternate coupling strategies, such as pulley waveguide couplers[15].

Next, we experimentally study $K_i$, $K_p$, and $K_s$ for a series of devices in which the resonator-waveguide gap is varied between 200 nm and 500 nm in Fig. 2(c). We observe the expected increase in $K_{i,p,s}$ with decreasing gap, that $K_i > K_p > K_s$ throughout, and that specific gap values can be chosen to target the high-performance overcoupled regime. For example, Fig. 2(d) shows the cavity mode transmission spectra at a gap of 300 nm. Of note are the high intrinsic quality factors achieved, e.g., $Q_{int} \approx \{5.2 \times 10^6, 6.1 \times 10^6, 3.3 \times 10^6\}$ for the signal, pump, and idler bands, respectively. This enables significant overcoupling to be achieved ($K_i \gtrsim 10$, $K_p \gtrsim 1$) while maintaining high overall $Q$s. In

comparison to other works utilizing resonators of a similar cross-section and size (i.e., an FSR of 1 THz)[16], the intrinsic $Q$s we observe are somewhat higher. This is likely a consequence of the relatively wide 3 μm ring widths we use, which limits the interaction of the optical field with the sidewalls. The ability to use such wide rings is a consequence of the hOPO scheme.

## High-performance OPO

We next characterize the hOPO performance of our dispersion- and coupling-engineered microresonators. Figure 3(a) shows the output spectrum when a device with a gap of 300 nm is pumped with $P_p = (87.3 \pm 3.0)$ mW of power at a frequency of 308 THz. As parasitic sidebands are suppressed, a well-isolated OPO spectrum with idler at 230 THz and signal at 386 THz is produced with an on-chip idler power of $P_i = (13.8 \pm 1.1)$ mW, corresponding to a conversion efficiency (in photon flux) $\eta_i \approx 0.21 \pm 0.03$ (the uncertainty values are described in the Fig. 3 caption). In contrast, traditional wide-span single-mode-family OPOs typically start to see the formation of parasitic sidebands at $P_i \approx 1$ mW. To elucidate the crucial role of resonator-waveguide coupling in the generation of high $\eta_i^{max}$ and high $P_i$ OPO, Fig. 3(b, c) plots these metrics as a function of resonator-waveguide gap,

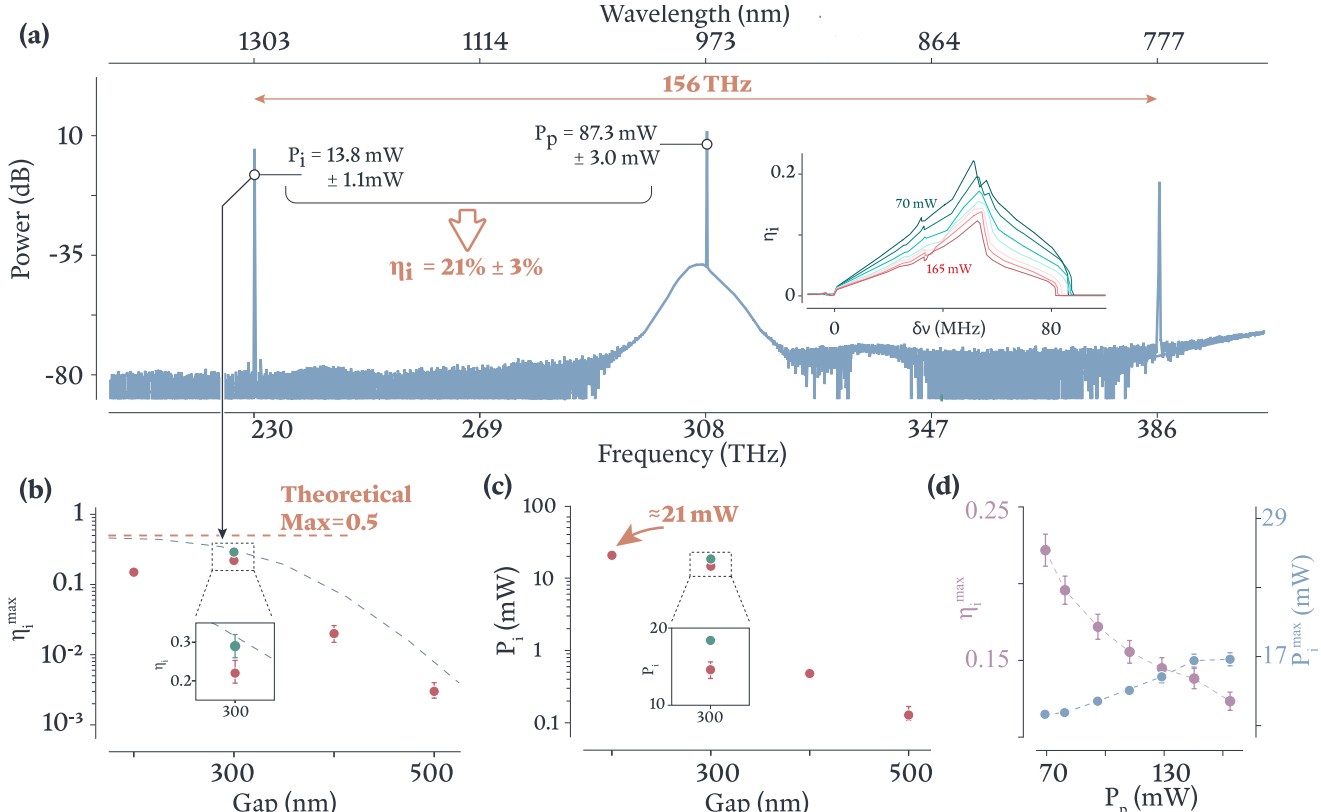

**Fig. 3 | High-performance OPO. a** Representative spectrum from an OPO device producing $(13.8 \pm 1.1)$ mW of on-chip idler power for an on-chip pump power of $(87.3 \pm 3.0)$ mW, corresponding to a conversion efficiency of $(0.21 \pm 0.03)$ (in terms of photon flux). A power of 0 dB is referenced to 1 mW (i.e., dBm); the offset between the plotted dB values and the on-chip power values $P_p$ and $P_i$ are due to fiber coupling and other insertion losses (see Supplementary Material for details). The inset displays $\eta_i$ and $P_i$ for different pump power (increasing from green to red) and pump frequency detuning. **b, c** Peak conversion efficiency $\eta_i^{max}$ (**b**) and on-chip idler power $P_i$ (**c**) as a function of resonator-waveguide gap for a series of OPO devices. $\eta_i^{max}$ and $P_i$ monotonically increase with decreasing gap (increased coupling) until a gap of 200 nm, at which point $\eta_i^{max}$ is reduced, as discussed in the text. The dashed gray line in **b** shows the theoretical value for $\eta_i^{max}$ based on the coupling parameters extracted in Fig. 2c, and the dashed orange line is the absolute theoretical maximum, where 50% of the pump photons are converted to the idler and the other 50% are converted to the signal. Two different data points are shown at a gap of 300 nm, corresponding to different OPOs. The OPO spectrum from the green data point in **c** is shown in Fig. 4(**b**) and represents the highest $\eta_i = 0.29 \pm 0.04$ with corresponding $P_i = (18.3 \pm 2)$ mW that we have observed. The uncertainty values and error bars in **b**–**d** are one standard deviation values and are due to the combined uncertainty of coupling losses on and off of the chip and the transmission losses of different optical components between the output coupling fiber and the detector (see Methods for details). For visual clarity, when the error bars are smaller than the data point size, they are omitted.

illustrating a strong increase in both as the resonator-waveguide gap decreases (and therefore as the $K_{p,i}$ increase; see Fig. 2(c)). $\eta_i^{max}$ based on Eq. (1) and the $K_{p,i}$ values from Fig. 2(c) are plotted as a dashed line in Fig. 3(b). In general, the measured $\eta_i^{max}$ values we report are still below the theoretical curve, likely due to imperfect frequency matching, where the pump detuning, Kerr, and thermal shifts do not perfectly compensate for $\Delta\nu$. In Fig. 3(c) we note that at smallest gap, we can generate isolated OPO with nearly 21 mW in the idler field (see Fig. 4(a)) despite some reduction in efficiency ($\approx$190 mW pump), likely due to the aforementioned imperfect frequency matching. Finally, Fig. 4(b) displays a spectrum with $\eta_i = 0.29 \pm 0.04$ and $P_i \approx 18.3$ mW, recorded in Fig. 3(b, c) as a green dot at a gap of 300 nm, which is one of the highest conversion efficiencies achievable in our devices. At these levels of $\eta_i$ and $P_i$, parasitic sidebands resurface, although the sidebands around the idler are still suppressed by more than 40 dB. Further improvements may be possible with continued device engineering to account for Kerr and thermal-shift contributions to the frequency mismatch.

We further consider the evolution of our OPO output as a function of pump power and pump laser tuning within the pump mode. In Fig. 3(a, inset) we plot a series of traces for waveguide-coupled pump powers between 70 mW and 165 mW, and for each trace we vary the pump laser frequency while recording $P_i$. The x-axis is referenced to the frequency at which the OPO threshold power ($\approx$30 mW) is dropped into the cavity. That is, for each waveguide-coupled pump power, we adjust the pump laser frequency until the dropped power in the cavity is high enough to reach threshold, and that value is taken as zero detuning. Beyond threshold, each trace displays a steady increase in $P_i$ (and hence $\eta_i$) until a maximum value for each is reached, after which the values decrease and eventually go to zero once frequency matching is completely lost. The detuning value at which this occurs depends on pump power, which is likely due to the differing thermal shifts of the cavity resonances and its impact on frequency mismatch. Once again, the strong normal dispersion for the pump mode family is successful in suppressing close-band parametric process, and pure OPO spectra with isolated pump/signal/idler tones like that in Fig. 3(a) are observed for much of the tuning range, though in some cases, the regions of highest conversion efficiency show additional parametric sidebands, as noted above and shown in Fig. 4(b). The additional sidebands remain strongly suppressed around the idler (40 dB to 50 dB below the idler), while in the signal band, they are only suppressed by $\approx$15 dB to 20 dB.

The absence of sidebands in Fig. 4(a) and their presence in Fig. 4(b) can be motivated from the understanding presented in the Suppressing Parasitic Processes section above. Namely, at 200 nm gap, the circulating power of each of the pump, signal,

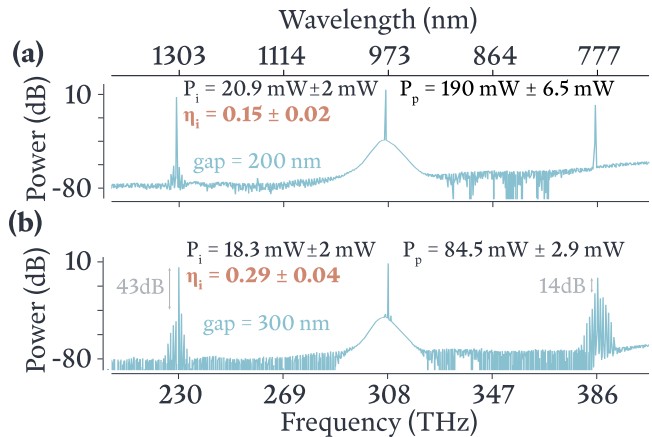

**Fig. 4 | Highest output power and conversion efficiency OPOs. a** Spectrum of the greatest output power OPO, which was achieved with a 200 nm gap and has $P_i$ beyond 20 mW, albeit at a conversion efficiency of $\eta_i \approx 0.14$. At high output powers, this device continues to show isolated OPO. The loaded Qs for the idler, pump, and signal are $6.1 \times 10^4$, $2 \times 10^5$, and $1.2 \times 10^6$, respectively, for the 200 nm coupling gap. **b** Spectrum of the greatest conversion efficiency OPO, which was achieved with a 300 nm gap and has $\eta_i$ approaching 30%, at the expense of additional parametric sidebands. The loaded Qs for the idler, pump, and signal are $3 \times 10^5$, $1.6 \times 10^6$, and $4 \times 10^6$, respectively, for the 300 nm coupling gap. In the idler band, the additional sidebands are > 40 dB below the idler, while in the signal band, the additional sidebands are > 14 dB below the signal.

and idler fields is lower than that at 300 nm gap since the 300 nm gap is closer to critical coupling. Since the geometry of the rings (and hence their dispersion) is nominally the same, the greater circulating power within the 300 nm gap ring supports sideband formation more strongly than the power within the 200 nm gap ring. Furthermore, the large spectral separation ( >150 THz) between the signal and the idler allows the two tones to experience vastly different levels of cavity enhancement. Thus, considering the 300 nm gap device alone, one may expect the greater cavity enhancement near the signal (see Fig. 2) to support sideband formation more strongly than that near the idler. Nevertheless, some ambiguity persists because of the possibility that mode couplings result in local regions of near-zero dispersion (e.g., seen in the spread in the dispersion data) that might promote parasitic processes once the signal and/or idler power become sufficiently strong, and these mode couplings may differ for devices with different waveguide coupling parameters.

Finally, Fig. 3(d) shows $\eta_i^{\max}$ and $P_i^{\max}$ for this OPO device as a function of waveguide-coupled pump power. We observe a saturation of $P_i$ and a monotonic reduction in $\eta_i$ as the pump power increases, likely due to the aforementioned combination of pump-power-dependent frequency mismatch (due to thermal shifts) and competing parasitic processes near the signal and idler bands as explained above. Further mitigation of these effects is needed to continue to improve the power performance and conversion efficiency of these devices.

## Discussion

In considering the potential applications and future development of the OPO devices we describe in this work, it is useful to place their performance in context with other demonstrated OPO systems[9,12,14,17–33]. Figure 5 illustrates the output power for various OPO demonstrations, ranging from chip-integrated technologies to mm-scale resonators to larger table-top technologies, as a function of their input power. While OPO has seen tremendous progress at all scales, it stands to reason that a widely deployable on-chip device should have access to an on-chip pump source and produce output power suitable

for downstream applications, which in some cases may require more than one milliwatt output on chip. As highlighted by the red gradients in Fig. 5, power considerations alone greatly reduce the number of OPO demonstrations amenable to deployable systems. The additional requirements of small-size and compatibility with silicon photonics leaves our microresonator device as a strong contender for the rapid and scalable deployment of a wide-wavelength-access laser system that simultaneously achieves high output power and conversion efficiency. In particular, we note the distinct application space for our chip-integrated system, where pump, signal, and idler waves are all widely separated in frequency, e.g., in comparison to nearly degenerate systems in either $\chi^{(2)}$ or $\chi^{(3)}$ platforms.

Furthermore, while Fig. 5 contextualizes the output power and conversion efficiency of our passively cooled device, other investigations[12,14] have noted that supplementing similar $\chi^{(3)}$ OPO systems with active thermal tuning can aid in aligning the OPO output wavelengths to an externally defined target wavelength. Given recent advancements[18] in high-efficiency thermal tuning solutions for on-chip nonlinear nanophotonics devices, we expect thermal control of the refractive index to further strengthen the suite of features afforded by high performance OPO.

We emphasize that while integrated microresonator OPOs have realized high efficiency previously, it has largely been in a regime of low output powers ($\approx$100 μW), with the exceptions of refs.[14,19], where a few mW of signal and idler power were generated from a 780 nm band pump, but the signal-idler separation was limited to a few tens of THz. Our work realizes output powers exceeding 20 mW at a signal-idler separation greater than 150 THz, while operating with pump powers that are still accessible from compact laser sources. This performance has been made possible through a combination of coupling engineering to efficiently inject the pump and extract the OPO output, and dispersion engineering to promote the nonlinear process of interest while suppressing competing parasitic processes. The former involves overcoupling of both the pump mode and targeted output mode (idler in our case) while maintaining high loaded $Q$, while the latter involves the use of a hybrid mode matching scheme in which all modes of interest are situated in regions of normal dispersion, but are nevertheless still able to realize phase- and frequency-matching. We note that table-top OPOs routinely produce significantly more than 100 mW of output power at high efficiency, though they require pump powers that are typically not easily available from compact laser sources. On the other end of the spectrum, breakthrough devices[20,34] continue to usher in ultra-low threshold powers, but thus far with limited output powers. We note that in general, ultra-low threshold and high output power are not achieved simultaneously in $\chi^{(3)}$ microresonator OPO devices, due to power-dependent Kerr frequency shifts. These shifts prevent a system that is frequency-matched for low power operation from being frequency-matched for high power operation, and degrading the device Q, to reduce the Kerr shifts by reducing the intracavity intensity, itself increases the threshold power.

To summarize, we demonstrate high-performance on-chip microresonator optical parametric oscillation that produces > 15 mW of output power at conversion efficiencies > 25%, without compromising on the span of the output signal and idler frequencies ( >150 THz signal-idler separation). Simultaneously realizing these three features in an on-chip OPO represents a significant advance in the realization of flexible wavelength access for lasers. Furthermore, its development on a platform compatible with silicon photonics makes it well suited for wide-scale deployment outside of laboratory settings. This was accomplished by suppressing competitive processes within the resonator through the use of an hOPO scheme and by engineering the coupling to an access waveguide. Going forward, we expect combined coupling engineering and flexible frequency matching techniques, such as the hybrid mode-matching scheme used in this work (or recently implemented photonic crystal microring approaches[35,36]),

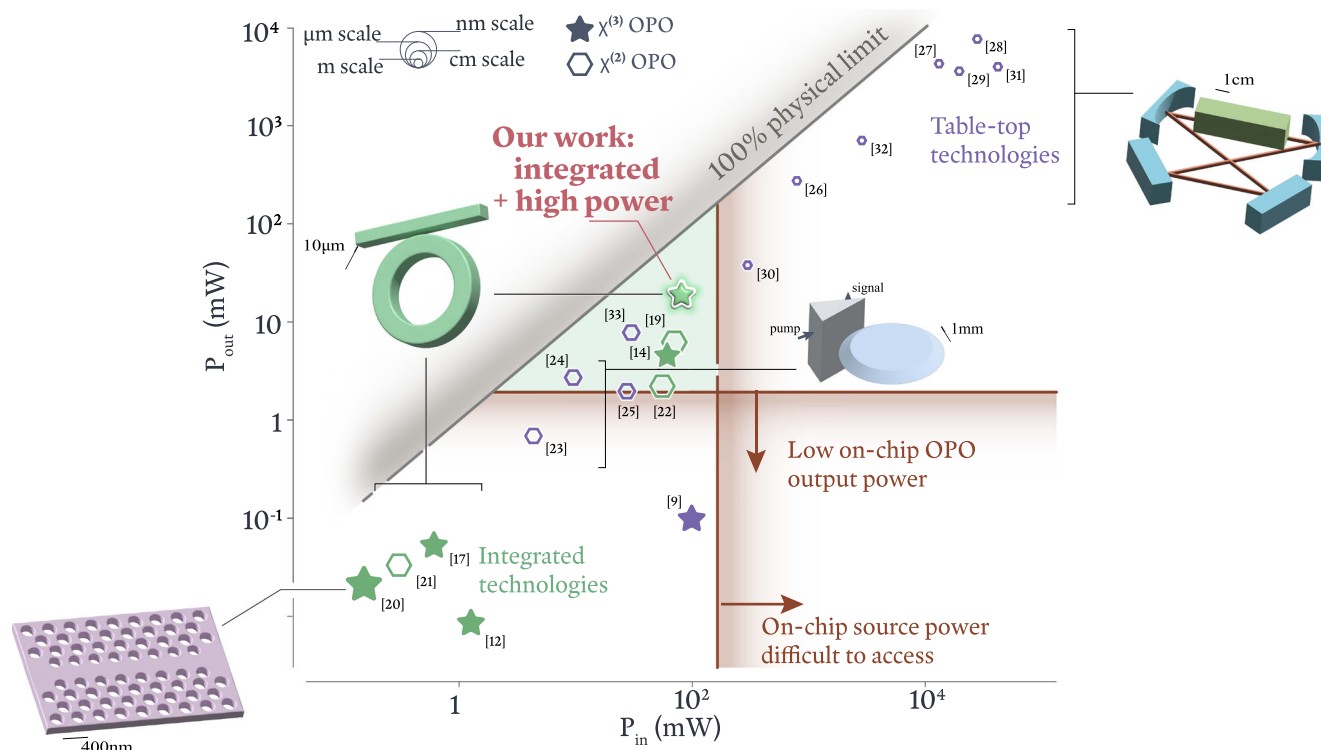

**Fig. 5 | OPO performance in context.** Output power vs. input power for different demonstrated OPO devices, including both integrated (green) and non-integrated (purple) technologies and both $\chi^{(3)}$ (stars) and $\chi^{(2)}$ (hexagons) nonlinearities. In general, integrated technologies (lower left) that have realized high conversion efficiency do so at low output powers, due to factors such as a large frequency mismatch (worsened by Kerr and thermal shifts depending on power, as described in the text) or inadequate suppression of competing processes at higher powers. On the other hand, large table-top technologies (upper right) have simultaneously realized high output powers and high conversion efficiencies, with the caveats of not being widely scalable, in part due to the requirement of pump powers beyond those easily available from compact lasers. Between the low and high power extremes lies an important regime (center, green shade) for deployable laser technology, where input powers are available on-chip and output powers are sufficient for many downstream applications. Our work demonstrates the capacity for $\chi^{(3)}$ silicon-photonics-based OPO to access the highlighted region of performance space, and is (to the best of our knowledge) the only integrated device in the region with wide wavelength separation between the pump, signal, and idler waves. The size of the data points (hexagons and stars) is inversely proportional to the device footprint, with the upper left circles providing a coarse scale bar.

to enable high-performance OPO across different wavelength bands, including the visible[37] and mid-infrared[9,38]. Such work would further establish microresonator OPO as a practical approach for realizing high-performance laser wavelength access across a broad spectral range.

## Methods
### Characterization of OPO devices
The OPO devices under study were characterized using the experimental setup shown in Supplementary Fig. S1. For linear characterization of the devices, continuously tunable lasers (CTL) in the 780, 980, and 1300 nm bands were used. Each laser has a piezo-controlled (PZ-controlled) fine-frequency sweep modulated by a signal generator whose voltage is recorded by a data acquisition device (Oscilloscope in drawing). At the output of each laser, cascaded 90/10 and 50/50 couplers (at the appropriate wavelengths) provide light for two frequency measurements. As the PZ control voltage is modulated and recorded, the absolute frequency of the light is measured at the peaks of the driving saw-tooth voltage using a wavemeter. In between the saw teeth, displacements from the wavemeter-referenced frequencies are continuously recorded by Mach-Zehnder interferometers (MZIs) that feed into wavelength-compatible photodiodes. The use of an MZI allows us to track nonlinear deviations in the frequency sweeps of the laser. The 90% tap of the lasers is then attenuated to low powers ($\approx 10\,\mu W$) using fiber-coupled attenuators (atten.) and aligned to a polarization of interest using fiber-coupled polarization controllers (PCs). Next, 10%

(nominal rating at 980 nm) of the light is tapped to a power meter ($PM_{in}$) to measure and monitor the power during subsequent steps and measurements. Finally, light is injected into the on-chip waveguide with a 980 nm lensed fiber. At the output of the chip, a similar lensed fiber is used to collect emission from the on-chip waveguide. 10 % of the collected light is routed to an optical spectrum analyzer (OSA) for analysis. Of the remaining light, 10% is tapped to monitor the insertion loss between points (A) and (B) in the diagram. To ensure precision, each of the couplers in the set up were characterized in each of the three wavelength bands. For simplicity, the nominal (i.e., 90/10 or 50/50) ratios are reported in the figure. The final portion of the light reaches a 980/1300 wavelength division multiplexer (WDM). The 980 nm light is coupled to a visible-light photo detector ($PD_{VIS}$). The 1300 nm path routes the light through an 1100 nm long pass filter before being coupled to an infrared photodiode. The voltage of either $PD_{VIS}$ or $PD_{IR}$ (depending on the laser utilized) is recorded simultaneously with the voltages from the frequency measurement (i.e., the signal generator, MZI, and wavemeter measurements). The recorded voltages are then processed to provide transmission scans, like those shown in Fig. 2(d), which are then fit to Lorentzian lineshapes to determine the Qs of various devices. The dispersion measurements of our devices use the same setup and are acquired by manually tuning the laser frequency to the bottom of a resonance and measuring its frequency using a wavemeter.

For OPO characterization, the same set up was employed without the green dashed-line bypass. In this scheme, only the 980 CTL is used as a pump laser, with its output amplified by a tapered amplifier. In this

configuration, the output photodiode ($PD_{IR}$) measures voltage $V_{IR} = \gamma_i P_i$. The value of $\gamma_i$ is determined by measuring the losses between (b) and (d) in the schematic, after which the measurements shown in Fig. 3 can be recorded.

## Geometric characterization of OPO devices

The geometry of the OPO devices was explicitly checked through the use of focused ion beam (FIB) milling, which allows one to image a cross section of the ring resonator. A representative cross section is shown in Supplementary Fig. S2, which resembles an upside-down trapezoid with sidewall angles of approximately 16 degrees, as indicated in the image. The notch at the top of the trapezoid is unintended and likely due to the reflow and chemical mechanical polishing processes used in fabrication. However, the notch has a limited impact on the device dispersion since the optical fields circulate near the edge of the device, away from the notch (see Fig. 2(a) insets).

## Data availability

The data that supports the plots within this paper and other findings of this study are available from the corresponding authors upon reasonable request.

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

## Acknowledgements

This work is supported by the DARPA LUMOS and NIST-on-a-chip programs. The authors thank Dr. Junyeob Song from NIST for assistance with focused ion beam characterization of the microring resonator cross-section.

## Author contributions

E.P. led the project, conducted the experiments, data analysis, and paper writing. G.M. contributed to theoretical understanding, experimental design, data collection, and microresonator design. F.Z., X.L., and K.S. contributed to the understanding and characterization of the hOPO scheme employed. J.S. and K.S. contributed to the understanding and execution of parasitic process suppression. E.P., G.M., and K.S. prepared the manuscript. All the authors contributed to and discussed the content of this manuscript.

## Competing interests

NIST/UMD have filed patent applications, with X.L. and K.S. listed as inventors, related to the work presented in this article. E.F.P., G.M., J.S., and F.Z. declare no competing interests.
