## [Peer review file · Nature Communications]

REVIEWER COMMENTS

Reviewer #1 (Remarks to the Author):

In this manuscript, authors demonstrate the operation of a ring resonator as an optical parametric oscillator where nearly ideal conversion efficiency into signal and idler is achieved. The key idea is to consider four wave mixing involving two "families" of modes (here TE and TM) both being with positive dispersion. While positive dispersion prevents FWM between any generic triplet of modes, the mode diversity enables modal phase matching of a single triplet of well-separated modes. This is extremely effective in preventing spurious interactions. Maximal efficiency is achieved by carefully tuning the coupling to the waveguide, as a trade off with OPO pump power at threshold (fig 3). An interesting point is also the achievement of very large power in the signal and idler, still preventing the emergence of parasitic sidebands (fig 4).

The manuscript is carefully prepared, well written, clear and so are the figures. Claims are well supported. The performances achieved are world-class, as well substantiated in fig 5.

I therefore recommend the publication, after Authors have given consideration to the following comments:

- controlling parasitic interaction is an issue of highly multimode resonators, such as ring resonator, whispering gallery modes, where essentially all modes are very similar and it is very difficult to select a certain subset of them. In fact, the properties of a large number of modes are easily described by the dispersion. if dispersion is small and anomalous, all modes tend to participate to four wave mixing and other parametric interactions. This does not happen in much smaller resonators. Thus, authors brilliantly add a new functionality in ring resonators (and similar optical cavities). In other kind of resonators, this is not needed because parasitic parametric effects are unlikely. Yet, ring resonators and similar are a very important technology...

- One claim left me surprised: " it stands to reasonproduce more than e.g. 1 mW output". Why 1 mW? how is this justified? what applications have in mind? Clearly there are applications needing a minimum amount of power, for instance for Shot noise limited detection. There are others where it is all about minimizing the total amount of power, and detecting a few photons is enough... so I would not agree with this. I suggest this statement is reconsidered, and perhaps a little change is made in figure 5, with a more nuanced, less "universal" reference to output power requirement. By no means this would diminish the importance of what authors have achieved.

Reviewer #2 (Remarks to the Author):

The authors have provided a complete, convincing, and highly technically impressive result demonstrating a crucial advance -- integrated OPOs that are high-power and highly efficient.

I have only minor comments and requests for clarification:

- Are all three modes coupled to the same mode of the coupling waveguide?
- Is the device tuned in some way (e.g., by temperature) to better align the modes spectrally? How would it benefit from such tuning if it is not? Does this device need to be operated in a thermally stabilized manner?
- More comments on the apparent rise of admittedly weak parasitic sidebands on both the signal and idler, as shown in Figure 4b) would be welcome. These sidebands appear at lower power and higher resonator Q when compared to 4a) (total Qs would be helpful). The authors hint at some possible explanations. Considering the importance of having pure tones in operating such an OPO, a bit more detail on experimental results and/or a deeper analysis of possible processes would be invaluable.

Happy to recommend this work for publication.

Reviewer #3 (Remarks to the Author):

The manuscript "High-performance microresonator optical parametric oscillator on a silicon chip" by Perez et al. presents the experimental demonstration of the optical hyperparametric oscillations by a silicon microring cavity. Although optical parametric oscillation and optical hyperparametric oscillations have been demonstrated on similar platforms for decades, the authors could achieve a very high conversion efficiency of up to 29% under a pump power of about 70mW. In this work, the authors show a design of the microring cavity by employing both TE and TM modes in the four-wave mixing process and demonstrate the suppression of spurious effects (e.g. comb generation) by using normal dispersion. By carefully designing the waveguide-microring coupling gap using a single bus waveguide, a relatively pure hyperparametric oscillation has been realized with on-chip pump powers above 70mW. Compared with previous related works, the authors claim a high power, which has never been achieved in integrated photonic devices.

However, the ideas behind the devices are not new, and the performance of the system might not be comparable with previous demonstrations concerning the thresholds. It is not clear what significant new design or new physics lead the authors to the high-power on-chip hyperparametric oscillator. As a result, I do not think the novelty of this paper reaches the level of Nature Communications.

Detailed comments:

1. Optical hyperparametric oscillations through four-wave mixing have been investigated intensively in nonlinear microcavities, which is different from the optical parametric oscillations through $\chi^{(2)}$ nonlinearity. I would suggest using a different name to distinguish the two processes.
2. It is a good idea to use the normal dispersion to suppress the unwanted four-wave mixing process. However, it is well-known that the microcomb generation requires abnormal dispersion. Therefore, it is not surprising to use normal dispersion to suppress the comb generation. In addition, for $\chi^{(2)}$ based OPO process, there is no such spurious effect.
3. The authors present many details about matching and coupling. It seems that phase matching is very sensitive and difficult to achieve in practice, so how could this device be applied in the future? In addition, it is not clear how to tune the lasing frequency, and what is the tuning range of the lasing wavelength, which is also critical in applications.
4. The authors have previously used two bus waveguides to separately couple idle and signal modes, why do not use that structure in this work? It seems that the performance of the device is limited by the single bus waveguide structure, as shown in Fig. 2(c).
5. For a lasing source, the threshold is critical. I can not find any unambiguous description of the threshold. Although the authors mention threshold several times when discussing the inset of Fig. 3(a), I could not understand how "the x-axis referenced to the OPO threshold".
6. In the past decades, there are tens of publications that demonstrate the OPO or hyperparametric oscillation with integrated photonic devices, and part of them are summarized in Fig. 5 by the authors. Most of the works are pursuing the low threshold OPO, and high-efficiency OPOs have already been demonstrated with mW or even sub-mW pump powers. It is obvious that if they reduce the quality factors of their microcavities, the threshold could be increased to 10-100mW, and then they can achieve high output power. So, I can not be convinced that the high output power is a challenge.

NCOMMS-22-37643 (“High-performance microresonator optical parametric oscillator on a silicon chip”) Response Letter

We thank all three referees for their constructive comments, which have helped us improve the presentation of our work. We provide a point-by-point response to the referee comments below in blue. In addition, the revisions we have made to our manuscript are in blue color.

REVIEWER COMMENTS

Reviewer #1 (Remarks to the Author):

In this manuscript, authors demonstrate the operation of a ring resonator as an optical parametric oscillator where nearly ideal conversion efficiency into signal and idler is achieved. The key idea is to consider four wave mixing involving two "families" of modes (here TE and TM) both being with positive dispersion. While positive dispersion prevents FWM between any generic triplet of modes, the mode diversity enables modal phase matching of a single triplet of well-separated modes. This is extremely effective in preventing spurious interactions. Maximal efficiency is achieved by carefully tuning the coupling to the waveguide, as a trade off with OPO pump power at threshold (fig 3). An interesting point is also the achievement of very large power in the signal and idler, still preventing the emergence of parasitic sidebands (fig 4).

The manuscript is carefully prepared, well written, clear and so are the figures. Claims are well supported. The performances achieved are world-class, as well substantiated in fig 5. I therefore recommend the publication, after Authors have given consideration to the following comments:

- controlling parasitic interaction is an issue of highly multimode resonators, such as ring resonator, whispering gallery modes, where essentially all modes are very similar and it is very difficult to select a certain subset of them. In fact, the properties of a large number of modes are easily described by the dispersion. if dispersion is small and anomalous, all modes tend to participate to four wave mixing and other parametric interactions. This does not happen in much smaller resonators. Thus, authors brilliantly add a new functionality in ring resonators (and similar optical cavities). In other kind of resonators, this is not needed because parasitic parametric effects are unlikely. Yet, ring resonators and similar are a very important technology...

We thank the reviewer for their positive assessment of our work. We agree that the simplicity of systems that are naturally free of parasitic processes (potentially due to being smaller cavities) is a very compelling feature, and indeed, small mode volume photonic crystal resonators, for example those in Ref. 19, are certainly an interesting platform for realizing such a system. Nevertheless, as the reviewer states, ring resonators are important technologies, and in particular for our application, provide access to high cavity quality factors across widely separated wavelengths. We are thus pleased that our work starts with a many-mode resonator and takes it to the limit where it behaves like a three mode system and can thus achieve high output power and high conversion efficiency without sacrificing broad wavelength access (>150 THz signal/idler separation), all while maintaining compatibility with silicon photonics fabrication technologies.

- One claim left me surprised: " it stands to reasonproduce more than e.g. 1 mW output". Why 1 mW? how is this justified? what applications have in mind? Clearly there are applications needing a minimum amount of power, for instance for Shot noise limited detection. There are others where it is all about minimizing the total amount of power, and detecting a few photons is enough... so I would not agree with this. I suggest this statement is reconsidered, and perhaps a little change is made in figure 5, with a more nuanced, less "universal" reference to output power requirement. By no means this would diminish the importance of what authors have achieved.

We appreciate the referee's point here - while some applications require more than 1 mW output power, there are certainly others which do not. Indeed, the 1 mW number was chosen as being representative of what one might typically find available from commercial compact laser systems. Such an output power is relevant in certain contexts, for example, a number of AMO-style experiments where the OPO might be a direct replacement for compact chip-based sources like DFB lasers.

To make it clear that this number is rather generically chosen, we now write "... it stands to reason that a widely-deployable on-chip device should have access to an on-chip pump source and produce output power suitable for downstream applications, which in some cases may require more than one milliwatt."

We have also modified Figure 5, as the referee suggested, to avoid any implication that there is some universal power requirement. Instead of writing 'OPO output power limits applications' for the region below 1 mW, we write 'low on chip OPO output power.'

Reviewer #2 (Remarks to the Author):

The authors have provided a complete, convincing, and highly technically impressive result demonstrating a crucial advance -- integrated OPOs that are high-power and highly efficient.

We thank reviewer #2 for their positive evaluation of our work and make point-by-point comments to each of the following questions.

I have only minor comments and requests for clarification:

- Are all three modes coupled to the same mode of the coupling waveguide?

We appreciate the reviewer's attention to this detail. We expect the modes to couple back into modes of the same polarization, meaning the ring's fundamental TE modes (pump, idler) would couple back to the waveguide's fundamental TE modes and the ring's fundamental TM mode (signal) would couple back to the waveguide's fundamental TM mode. Regarding higher-order transverse spatial modes (not used in this study), fundamental modes in the waveguide can couple to higher-order modes in the ring and vice-versa, which is a process that should be considered during waveguide design.

- Is the device tuned in some way (e.g., by temperature) to better align the modes spectrally? How would it benefit from such tuning if it is not? Does this device need to be operated in a thermally stabilized manner?

In this work, the on-chip device's temperature is not externally managed. The device is passively cooled in an ambient-temperature room. The following details regarding this aspect of the experiment were added to the end of the first paragraph in the section entitled "Device design, dispersion, and coupling"

"Finally, the temperature of the devices was not actively managed; they were passively cooled in an ambient-temperature room."

Thus, the device is not externally temperature-tuned, but simply cooled by room-temperature air in the lab. Nevertheless, the use of external temperature tuning in similar systems has been studied in the past [Ref. 12, Ref. 14] and is beneficial in two ways. As discussed in the text, in the current investigation, phase and frequency matching is achieved by controlling the ring's geometry, but temperature tuning the material could supplement this through the thermo-optic effect. As an additional benefit, temperature tuning can also be used to shift the three tones that are frequency matched into alignment with an external wavelength target (typically this would be done through a combination of temperature tuning and adjustment of the pump frequency). However, we note that the sensitivity to temperature is a function of the device dispersion that needs to be considered in device design.

While some of these details were included in the first paragraph of the section entitled "Suppressing parasitic process" the following passage was added after the first paragraph of the Discussion and Context section:

Furthermore, while Fig.5 contextualizes the output power and conversion efficiency of our passively cooled device, other investigations (Ref.12, Ref.14) have noted that supplementing similar $\chi^{(3)}$ OPO systems with active thermal tuning can aid in aligning the OPO output wavelengths to an externally defined target wavelength. Given recent advancements (ref. 34) in thermal tuning solutions for on-chip devices, we expect thermal control of the refractive index to further strengthen the suite of features afforded by high performance OPO.

- More comments on the apparent rise of admittedly weak parasitic sidebands on both the signal and idler, as shown in Figure 4b) would be welcome. These sidebands appear at lower power and higher resonator Q when compared to 4a) (total Qs would be helpful). The authors hint at some possible explanations. Considering the importance of having pure tones in operating such an OPO, a bit more detail on experimental results and/or a deeper analysis of possible processes would be invaluable.

We thank the referee for their thoughtful questions, whose answers add value to the work.

In general, the presence of parasitic sidebands depends on several factors, including (1) phase- and frequency-matching of the parasitic process; (2) cavity enhancement of the parasitic process; and (3) the difference in the coupling regime (overcoupled / undercoupled) for each of the pump signal and idler waves, as quantified by the coupling parameter K (>1 for overcoupled and <1 for undercoupled).

Our work suppresses parasitic processes to achieve pure tones through the use of strong normal dispersion, as theoretically proposed in Ref. 10, where the rough takeaway is that the more strongly-normal the dispersion is near the pump, the more circulating power the resonator can support without creating sidebands. Since we focus on a fixed ring-resonator geometry in this work (only the resonator-waveguide gap is changing), the first factor above shouldn't change from device to device, besides the change in linewidth associated with change in gap, while factors 2 and 3 do change. Our understanding is that the cavity enhancement, in particular, is the critical factor in our experiments.

At 200 nm gap, where the cavity is overcoupled at all of the relevant wavelengths, the cavity enhancement, and thus the intra-cavity power, is small (compared to critical coupling), which suppresses the creation of parasitic sidebands as explained by the rough takeaway from Ref.10. On the other hand, at 300 nm gap, the intra-cavity power of each wavelength is greater than that in the 200 nm gap design, which increases the likelihood of sideband formation. To answer the reviewer's question, then, the increased cavity enhancement in the 300 nm gap design (compared to the 200 nm gap) is itself what leads to parasitic processes being generated for a smaller input power than they would for the 200 nm gap design.

Finally, at all gaps the coupling parameter increases with wavelength ($K_i > K_p > K_s$). So, in the 300nm gap design, the larger sideband around the signal may be due to the fact that the intracavity power of the signal is two orders of magnitude higher than that of the idler, which itself is due to the large frequency separation (>150THz) between the signal and idler.

Ultimately, it's likely that a thorough theoretical investigation, similar to that of Ref. 10, is needed to detangle the chronological development of the sideband formation. However, we fully agree with the reviewer that this is an important aspect of the work, and are therefore addressing it through some additions to the main text, where we explicitly restate some notable pieces of information (most of which were already elsewhere in the investigation). The following passage has been added as the third paragraph of the High-performance OPO section;

“The absence of sidebands in Fig. 4(a) and their presence in Fig. 4(b) can be motivated from the understanding presented in the Suppressing Parasitic Processes section above. Namely, at 200~nm gap, the circulating power of each of the pump, signal, and idler fields is lower than that at 300~nm gap since the 300~nm gap is closer to critical coupling. Since the geometry of the rings (and hence their dispersion) is nominally the same, the greater circulating power within the 300~nm gap ring supports sideband formation more strongly than the power within the 200~nm gap ring. Furthermore, the large spectral separation (>150 THz) between the signal and the idler allows the two tones to experience vastly different levels of cavity enhancement. Thus, considering the 300 nm device alone, one may expect the greater cavity enhancement near the signal (see Fig.~\ref{fig:2}) to support sideband formation more strongly than that near the idler. Never the less, ambiguity persists because of the possibility that mode couplings result in local regions of near-zero dispersion (e.g., seen in the spread in the dispersion data) that might promote parasitic processes once the signal and/or idler power become sufficiently strong.”

We have also added the $Q_{\{t\}}$ information to Figure 4 as requested by the reviewer

Happy to recommend this work for publication.

We appreciate the reviewer's recommendation of our work.

Reviewer #3 (Remarks to the Author):

The manuscript "High-performance microresonator optical parametric oscillator on a silicon chip" by Perez et al. presents the experimental demonstration of the optical hyperparametric oscillations by a silicon microring cavity. Although optical parametric oscillation and optical hyperparametric oscillations

have been demonstrated on similar platforms for decades, the authors could achieve a very high conversion efficiency of up to 29% under a pump power of about 70mW. In this work, the authors show a design of the microring cavity by employing both TE and TM modes in the four-wave mixing process and demonstrate the suppression of spurious effects (e.g. comb generation) by using normal dispersion. By carefully designing the waveguide-microring coupling gap using a single bus waveguide, a relatively pure hyperparametric oscillation has been realized with on-chip pump powers above 70mW. Compared with previous related works, the authors claim a high power, which has never been achieved in integrated photonic devices.

However, the ideas behind the devices are not new, and the performance of the system might not be comparable with previous demonstrations concerning the thresholds. It is not clear what significant new design or new physics lead the authors to the high-power on-chip hyperparametric oscillator. As a result, I do not think the novelty of this paper reaches the level of Nature Communications.

We appreciate that the reviewer acknowledges the high conversion efficiency and output power we demonstrate while suppressing spurious effects, all within the context of an integrated photonics platform. We also appreciate the opportunity to discuss the novel aspects of our work - as well as the quantitative improvement in performance relative to earlier demonstrations of widely-separated OPO within a $\chi^{(3)}$ platform - and hope that this will address the reviewer's concerns regarding novelty.

Regarding threshold, we note that in a microresonator OPO, it is incongruous to simultaneously achieve the lowest possible threshold, high output power, and high conversion efficiency within the same device. This is simply because of Kerr nonlinear frequency shifts - a device that is optimized for low-threshold will experience limited Kerr shifts of its modes, and hence the cold-cavity frequency mismatch must be designed to be small. However, this means that at sufficiently high pump power, the system will become frequency mismatched and suffer a significant loss of efficiency. This has been clearly discussed in earlier works, e.g., Refs. 9 and 10. Beyond Kerr shifts, we note that for lowest threshold, out-coupling efficiency is not necessarily optimized, so devices can operate in an undercoupled regime in which loaded quality factors are higher and hence thresholds are lower. This has in fact been the case in our previous work [Ref. 12] where milliWatt thresholds were obtained, but output powers were in the tens of microWatt range at best.

To better explain our choice of metrics, we have added new text in the second paragraph of the 'Requirements for high performance' section.

Detailed comments:

1. Optical hyperparametric oscillations through four-wave mixing have been investigated intensively in nonlinear microcavities, which is different from the optical parametric oscillations through $\chi^{(2)}$ nonlinearity. I would suggest using a different name to distinguish the two processes.

We appreciate the desire to provide specificity with respect to which nonlinearity we utilize. We believe that the cleanest way to do so is to change the title of our manuscript to 'High-performance Kerr microresonator optical parametric oscillator on a silicon chip', where the use of 'Kerr' makes it clear that our work is based on the $\chi^{(3)}$ nonlinearity.

While we acknowledge that the term 'optical hyper-parametric oscillation' has also been used in some important investigations of $\chi^{(3)}$ systems [Matsko et al, *Phys. Rev. A* **71**, 033804 (2005); Razzari, et. al.,

Nature Photon **4**, 41-45 (2010)], it is by no means exclusively used. For example, very early work on $\chi(3)$ systems [Kippenberg, et. al., *Phys. Rev. Lett.* **93**, 083904 (2004)], as well as more recent landmark results [Sayson, et al, *Nature Photon* **10** (2019) and Marty, et al, *Nature Photon* **15** (2021)] all use the term ‘optical parametric oscillator.’ Moreover, we note the terminology of ‘hyperparametric oscillations’ has, in some cases, referred to close-band OPO processes based on modulation instability, for example, Matsko et al [*Phys Rev A* **71**, 033804, 2005] write that hyperparametric oscillations are ‘also known in fiber optics as modulation instability’ and ‘have phase matching conditions involving nearly degenerate optical frequencies.’ Neither is the case in our work.

We further note that in a principal text [Agrawal, *Nonlinear Fiber Optics*, Chapter 10.6.1], optical parametric oscillators are defined in the following:

“Perhaps, the simplest application of the parametric gain is to use it for making a laser by placing the fiber inside an optical cavity and pumping it with a single pump source... As a result, the laser emits the signal and idler beams simultaneously at frequencies that are located symmetrically on opposite sides of the pump frequency. Such lasers are called parametric oscillators;”

Given the important differences between our work and other $\chi(3)$ systems that use the ‘hyperparametric’ term (namely, operation outside of the anomalous dispersion/MI regime and with highly non-degenerate signal/idler/pump modes) and the functional similarities with $\chi(2)$ systems (including the direct comparison we provide in Figure 5), while also recognizing the importance of distinguishing between $\chi(2)$ and $\chi(3)$ processes, we think that the term “Kerr optical parametric oscillator’ is the best description of our work.

2. It is a good idea to use the normal dispersion to suppress the unwanted four-wave mixing process. However, it is well-known that the microcomb generation requires abnormal dispersion. Therefore, it is not surprising to use normal dispersion to suppress the comb generation. In addition, for $\chi(2)$ based OPO process, there is no such spurious effect.

We appreciate the reviewer’s understanding of the various effects at play (i.e., comb generation from abnormal dispersion, etc.), and agree that it is not surprising to use normal dispersion to suppress comb formation.

However, achieving sufficiently strong normal dispersion to suppress comb formation while simultaneously being able to phase and frequency match widely separated modes (> 150 THz in our work) is highly non-trivial. To phase and frequency match widely separated modes, many works [Lu *et al.*, *Optica* **7** (2020); Sayson *et al.* *Opt. Lett* **42** (2017)] rely on modes from the same transverse mode family, where higher-order dispersion is needed to balance the normal second-order dispersion. This dispersion balance tends to both be very sensitive to geometry and generally found in systems for which the second-order dispersion is only weakly normal. This weak normal dispersion can still result in modulation instability based parasitic processes, as discussed extensively in our recent theoretical work [Stone *et al.*, *Phys. Rev. Applied* **17**, 024038 (2022)].

Thus, the surprise in our work is not that strongly normal dispersion suppresses comb formation, but the subsequent high-performance device that we report *despite* the use of strongly normal dispersion. This is possible through the use of multiple mode families (what we refer to as a hybrid mode matching

scheme), which enables all modes of interest to be in regions of normal dispersion while still enabling their phase and frequency-matching.

We agree that the absence of spurious effects is a very compelling feature of $\chi(2)$ systems and we are pleased that the $\chi(3)$ system presented in this work can mimic such a compelling feature. We also note that $\chi(3)$ systems have their own compelling rationale, including direct compatibility with the popular silicon photonics platform that is widely available, including in foundries, and the fundamental fact that for a $\chi(2)$ system to reach the same output frequencies as we reach in our work, the pump would be situated at ~ 466 nm (half the wavelength we pump at). This poses a significant challenge for both dispersion engineering and broadband loss considerations, and to our knowledge, there is no existing chip-integrated $\chi(2)$ platform that has been able to realize OPO between such broadly separated frequencies.

To better clarify the above situation in our manuscript, we have made the following addition to the 'Discussion and context' section, second paragraph :

'This performance has been made possible through a combination of coupling engineering to efficiently inject the pump and extract the OPO output, and dispersion engineering to promote the nonlinear process of interest while suppressing competing parasitic processes. The former involves overcoupling of both the pump mode and targeted output mode (idler in our case) while maintaining high loaded Q , while the latter involves the use of a hybrid mode matching scheme in which all modes of interest are situated in regions of normal dispersion, but are nevertheless still able to realize phase- and frequency-matching.'

3. The authors present many details about matching and coupling. It seems that phase matching is very sensitive and difficult to achieve in practice, so how could this device be applied in the future? In addition, it is not clear how to tune the lasing frequency, and what is the tuning range of the lasing wavelength, which is also critical in applications.

We agree with the reviewer that there are many details about phase- and frequency-matching and coupling in this work, and that an understanding of these details is needed to realize high performance.

That being said, now that we have developed a basic prescription for realizing high-performance, we don't believe that reproducing our results will be especially difficult - at least, no more difficult than reproducing results for microresonator frequency combs and other nonlinear nanophotonics technologies being widely pursued by the integrated photonics communities.

To start with coupling, we note that the main requirements are to be overcoupled at the pump wavelength and even more strongly overcoupled at the output wavelength of interest (in this case the idler). This puts a premium on high intrinsic quality factor, which fortunately, is a quantity that the community is quite interested in and for which many important improvements have been demonstrated (see for example, the recent review article by Lipson's group, X. Ji et al, *APL Photonics*, **6**, 071101, 2021). With intrinsic Q s in the several million range now available in Si₃N₄ photonics, overcoupling by a factor of 10 is possible while retaining high loaded Q , which ensures that cavity enhancement is still significant.

Regarding phase- and frequency-matching while suppressing parasitic processes, as we noted in the response to the previous question, the scheme is paramount. Use of the hybrid mode family approach

not only enables these requirements, it does so robustly - in fact, in earlier work (Ref. 14), we showed that the robustness of the strategy to dimensional variation is much greater than widely-separated chi(3) OPOs based on a single mode family. Thus, we do not think that the phase-matching is in fact particularly sensitive or difficult to achieve in practice, provided that control of resonator geometry consistent with modern accurate nanofabrication methods is available.

Finally, we note that tuning of the OPO output frequency can occur through two mechanisms. The first is a very fine tuning provided by tuning the pump frequency within one linewidth, which modestly shifts the signal and idler frequencies. The second tuning knob uses external heaters to shift the signal, idler, and pump mode frequencies, which allows more extensive pump frequency tuning (and hence signal and idler output frequency tuning). As this tuning approach is not unique to our OPO work, we have chosen to add the following brief comments as the second paragraph in the "Discussion and context" section of our work:

"Furthermore, while Fig.5 contextualizes the output power and conversion efficiency of our passively cooled device, other investigations (Ref.12, Ref.14) have noted that supplementing similar chi-(3) OPO systems with active thermal tuning can aid in aligning the OPO output wavelengths to an externally defined target wavelength. Given recent advancements (ref. 34) in thermal tuning solutions for on-chip devices, we expect thermal control of the refractive index to further strengthen the suite of features afforded by high performance OPO."

To summarize, the robustness of our phase- and frequency-matching strategy with respect to geometry variations, in conjunction with frequency tuning capability, speaks to the usability of our OPOs in applications requiring high power.

4. The authors have previously used two bus waveguides to separately couple idle and signal modes, why do not use that structure in this work? It seems that the performance of the device is limited by the single bus waveguide structure, as shown in Fig. 2(c).

We appreciate the reviewer's awareness of related work from our group. The choice of waveguide coupling strategy was done with the application in mind - in this case, we wanted to realize a high-performance, widely-separated chi(3) OPO, and knew that one prerequisite was overcoupling of the pump and idler modes, with stronger overcoupling needed for the idler (since we focused on the idler mode as the output). Straight waveguide coupling provides a natural approach for doing so, because it couples more strongly at longer wavelengths, so that if the pump is slightly overcoupled, the idler can be strongly overcoupled. Thus, the performance in our work is not limited by straight waveguide coupling.

Of course, the situation changes if the higher frequency signal is the desired output, or if both the signal output and idler output are desired. In that case, the multiple bus waveguide geometries, use of pulley couplers, etc, would be preferable.

We address the latter point in the manuscript by writing (in the paragraph where the straight waveguide coupling approach is introduced:

'We note that other potential OPO objectives, such as high-efficiency and output power for the signal field, or equal efficiency and output power for both signal and idler fields, may require alternate coupling strategies, such as pulley waveguide couplers [16].'

5. For a lasing source, the threshold is critical. I can not find any unambiguous description of the threshold. Although the authors mention threshold several times when discussing the inset of Fig. 3(a), I could not understand how "the x-axis referenced to the OPO threshold".

We appreciate the referee's attention to this detail, which might be valuable for the readers. In the latest version of the manuscript, the value of approximately 30 mW is reported in the main text and in the caption of Fig. 3(a).

To clarify further, the inset to Figure 3a has an x-axis that is in units of frequency detuning. The zero value of frequency detuning is the point at which threshold is reached, that is, the pump frequency at which a sufficient amount of dropped power in the cavity to reach threshold (and corresponding to a waveguide input power of 30 mW) is achieved. We have adjusted the second paragraph of the "High-performance OPO section" accordingly.

6. In the past decades, there are tens of publications that demonstrate the OPO or hyperparametric oscillation with integrated photonic devices, and part of them are summarized in Fig. 5 by the authors. Most of the works are pursuing the low threshold OPO, and high-efficiency OPOs have already been demonstrated with mW or even sub-mW pump powers. It is obvious that if they reduce the quality factors of their microcavities, the threshold could be increased to 10-100mW, and then they can achieve high output power. So, I can not be convinced that the high output power is a challenge.

To our understanding, there are two parts to this comment. One regards the novelty of our results, e.g., where they stand within the landscape of related results, and the other regards the difficulty/challenge in achieving these results.

While low-thresholds have been pursued by many groups as the referee has noted (including ours in Ref. 12), we believe that the application space for higher output powers, e.g. in the mW and tens of mW range as we demonstrate in this work, is distinct and important. Simply put, there are numerous applications in AMO physics, communications, quantum science, spectroscopy, etc., that can not be serviced by sub-mW output powers, which will generally be the case for systems optimized for low threshold. As we have noted earlier in the response letter, such microresonator OPOs will generally not operate at higher output powers due to Kerr frequency shifts that lead to frequency mismatch, as well as unoptimized output coupling.

We believe that Figure 5 presents much of the needed context of where our work fits in with the existing state-of-the-art, in considering both $\chi(2)$ and $\chi(3)$ systems and chip-integrated, mm-scale, and table-top platforms. If the referee feels that we are missing some relevant contribution from the existing literature in Figure 5, we would be happy to learn which works we have neglected so that we can add them to the figure. That being said, we recognize that it is important to clarify in what sense our work is exceptional, and so we have introduced new text to the Figure 5 caption and accompanying main text.

In the main text, we now write 'The additional requirements of small-size and compatibility with silicon photonics leaves our microresonator device as a strong contender for the rapid and scalable deployment of a wide-wavelength-access laser system that simultaneously achieves high output power

and conversion efficiency. In particular, we note the distinct application space for our chip-integrated system, where pump, signal, and idler waves are all widely separated in frequency, e.g., in comparison to nearly degenerate systems in either $\chi(2)$ or $\chi(3)$ platforms.'

Finally, the referee's comment that high output is straightforward to achieve by, for example, reducing quality factor, is generally not true in practice for a widely-separated $\chi(3)$ OPO. That is because these OPOs are created in resonators (typically microrings in integrated photonics) that support many more modes than the three targeted modes for widely-separated OPO. Degrading the Q can lead to situations where these other modes participate in parasitic nonlinear processes because, in particular, the larger cavity linewidths enable frequency matching to be more easily achieved. In our system, it is only because the parasitic processes have been suppressed that it is possible to sustain 3 tone purity in the strongly over coupled (degraded Q) regime without sacrificing conversion efficiency.

Finally, as discussed in the "Increasing maximum conversion efficiency" section of the paper, and throughout the literature [Beckmann *et al.*, *Opt. Lett.* **37** (2012); Breunig *et al.*, *Laser Photonics Rev.* **10** (2016); Sayson *et al.*, *Nature Photon* **10** (2019), Stone *et al.* *Phys. Rev. Applied* **17** (2022)], high conversion efficiency depends on being strongly over coupled to the resonator, which becomes harder and harder to achieve as the resonator's intrinsic Q is degraded. Thus, at the device level, one should aim for:

1. As high intrinsic Q as possible
2. As low extrinsic Q as possible

As noted earlier, fortunately this scenario is becoming more feasible in silicon nitride integrated photonics, but it is due to recent development in the field, and has not been applied (prior to our work) to the widely-separated OPO problem (and as noted earlier, needs to be applied in conjunction with a phase and frequency-matching scheme that suppresses competing processes).

The above has been summarized in our 'Discussion and context' section with the following sentences:

'This performance has been made possible through a combination of coupling engineering to efficiently inject the pump and extract the OPO output, and dispersion engineering to promote the nonlinear process of interest while suppressing competing parasitic processes. The former involves overcoupling of both the pump mode and targeted output mode (idler in our case) while maintaining high loaded Q , while the latter involves the use of a hybrid mode matching scheme in which all modes of interest are situated in regions of normal dispersion, but are nevertheless still able to realize phase- and frequency-matching.'

REVIEWERS' COMMENTS

Reviewer #1 (Remarks to the Author):

I am fully satisfied with the reply of the Authors and, again, I congratulate for the excellent work carried out.

Reviewer #2 (Remarks to the Author):

The authors have answered my questions satisfactorily.

I would recommend publication.

Reviewer #3 (Remarks to the Author):

The authors have addressed most of my concerns and have revised their manuscript to clarify the novelty and advantage of their work. Now, I support the publication of this high-power Kerr OPO of microresonator in Nature Communications.